# The microbiome of the human facial skin is unique compared to that of other hominids

Samuel Degregori,[1] Melissa B. Manus,[1,2] Evan B. Qu,[3,4] Calen P. Mendall,[3,4] Jacob S. Baker,[3,4] Lydia M. Hopper,[5] Katherine R. Amato,[1] Tami D. Lieberman[3,4]

**ABSTRACT** The human facial skin microbiome is remarkably similar across all people sampled to date, dominated by facultative anaerobe *Cutibacterium*. The origin of this genus is unknown, with no close relatives currently described from samples of primate skin. This apparent human-specific bacterial taxon could reflect the unique nature of human skin, which is significantly more oily than that of our closest primate relatives. However, previous studies have not sampled the facial skin microbiome of our closest primates. Here, we profiled the skin microbiome of zoo-housed chimpanzees (*Pan troglodytes*) and gorillas (*Gorilla gorilla gorilla*), alongside their human care staff, using both 16S and shotgun sequencing. We showed that facial skin microbiomes differ significantly across host species, with humans having the lowest diversity and the most unique community among the three species. We were unable to find a close relative of *Cutibacterium* on either chimpanzee or gorilla facial skin, consistent with human specificity. Hominid skin microbiome functional profiles were more functionally similar compared to their taxonomic profiles. However, we still found notable functional differences, including lower proportions of fatty acid biosynthesis in humans, consistent with microbes' reliance on host-derived lipids. Our study highlights the uniqueness of the human facial skin microbiome and supports a horizontal acquisition of its dominant resident from a yet unknown source.

**IMPORTANCE** Understanding how and why human skin bacteria differ from our closest animal relatives provides crucial insights into human evolution and health. While we have known that human facial skin hosts distinct bacteria—particularly *Cutibacterium acnes*—we did not know if these bacteria and their associated genes were also present on the faces of our closest relatives, chimpanzees and gorillas. Our study shows that human facial skin hosts markedly different bacteria than other primates, with *C. acnes* being uniquely abundant on human faces. This finding suggests that this key bacterial species may have adapted specifically to human skin, which produces more oils than other primates.

**KEYWORDS** skin microbiome, comparative studies, primate, shotgun sequencing, 16S RNA

**Peer Reviewer** Holger Brüggemann, Aarhus University, Aarhus, Denmark

Address correspondence to Tami D. Lieberman, tami@mit.edu, or Katherine R. Amato, katherine.amato@northwestern.edu.

The authors declare no conflict of interest.

See the funding table on p. 9.

Among human skin sites, human facial skin harbors the highest density of bacteria, with comparable colony-forming units found only in the groin and axilla (1). The composition of this community is remarkably similar across all healthy adults sampled to date (all in industrialized contexts), dominated by the species *Cutibacterium acnes* and other members of the *Cutibacterium* genus (2–4). These facultative anaerobes live at high density in sebaceous follicles (hair follicles with large glands that produce lipid-rich sebum (5) and constitute ~70% of the relative abundance of facial skin microbes of adults (1–4). To date, no close relative of *Cutibacterium* has been reported on other

animals, making the origin of this species mysterious; the closest relatives described have been isolated from the rumen of cows and Swiss cheese (6). While the skin microbiomes of diverse mammals have been profiled (7, 8), studies of facial and other sebaceous skin sites are lacking. As *C. acnes* has many genes for metabolizing sebaceous lipids (9) and humans produce considerable amounts of sebum on their face and throughout their skin, with a substantially unique lipid profile (10, 11), it is possible that *C. acnes* may be specific to the human skin environment. However, non-human hominid species also have largely hairless faces with high densities of sebaceous glands relative to the rest of their bodies (12). Thus, *C. acnes* may have co-diversified with primates beyond just humans and may inhabit the facial skin of our closest relatives. Alternatively, *C. acnes* may have been acquired independently during human evolution and be absent from the facial skin microbiome (FSM) of other host relatives.

Here, to understand if *C. acnes* has a close relative on related hominids and to more generally understand how the human FSM differs from other hominids, we profiled the FSM of four chimpanzees and four gorillas at Lincoln Park Zoo in Chicago, IL, USA using both 16S rRNA and shotgun metagenomics. We also sampled four humans who were the apes' primary care staff. We used V1-V3 16S primers for 16S analysis, as *Cutibacterium* amplifies poorly using traditional V4 primers (13). Family-level composition is shown in Fig. 1A. Human FSMs exhibited significantly less alpha diversity, at the amplicon sequence variant (ASV) level, than gorilla and chimpanzee FSMs ($P_{adj} < 0.05$ Kruskal-Wallis; Fig. 1B), echoing results from other body sites (7). As expected, this decrease in diversity was associated with a dominance of *Propionibacteriaceae* (of which *C. acnes* is a member) and *Staphylococcaceae* in human FSMs (averaging 85% of these communities; Fig. 1B).

While all three hominid species had substantial proportions of *Staphylococcaceae* and *Mycobacteriaceae* (a family that includes *Corynebacteria*), they each had distinct FSMs—including significantly distinct communities on gorillas and chimpanzees ($P_{PERMANOVA} = 0.001$, Table S1). Among many differences, chimpanzee FSMs had a significantly higher relative abundance of *Mycobacteriaceae*, while gorilla FSMs had significantly high relative abundance of *Neisseriaceae* and *Lactobacillaceae* (Table S3, $P_{adj} < 0.04$). Notably, *Propionibacteriaceae* dominated human skin microbiomes, and only low abundances of *Propionibacteriaceae* were found on both chimpanzees and gorillas (0.013% and >0.001% relative abundance, respectively), aligning with a model in which *Cutibacterium* has recently adapted to thrive specifically in the oily skin of humans.

To investigate if the low abundance of *Propionibacteriaceae* on chimpanzees and gorillas might contain a close relative *C. acnes*—therefore supporting vertical diversification and bacterial evolution on hominids (14)—we performed several analyses. First, a phylogenetic analysis of prominent amplicon sequences assigned to *Propionibacteriaceae* revealed the absence of any non-human primate (NHP)-specific ASVs that could serve as a relative of the apparently human-specific *Cutibacterium* (Fig. 1C, left). In contrast, a wide range of closely related *Staphylococci* were found across human and NHPs at a high relative abundance (Fig. 1C, right), with several species specific to given hosts, implying a possible diversification from a recent common ancestor.

A low percentage of 16S and metagenomic reads from the NHPs were classified as *C. acnes*. This could reflect the transmission of *C. acnes* from the humans to NHPs in the zoo or contamination of NHP samples in the lab, or it might reflect a close relative of *C. acnes* indistinguishable using 16S. To distinguish between these possibilities, we compared metagenomic reads to a phylogeny-directed reference database containing publicly available *C. acnes* genomes, all collected from humans. Our analysis concluded that the trace amounts of *C. acnes* on NHPs were not more significantly diverged than those on the four people sampled, and overall, neither group had strains that were highly diverged from known *C. acnes* genomes (Fig. 1D). Moreover, attempts to culture *C. acnes* from these NHP samples were unsuccessful (Methods, Table S9). These results suggest that the trace amounts of *C. acnes* in NHP samples have a direct human origin. Lastly, direct assembly from metagenomes did not return any contigs suggestive of a *C. acnes*

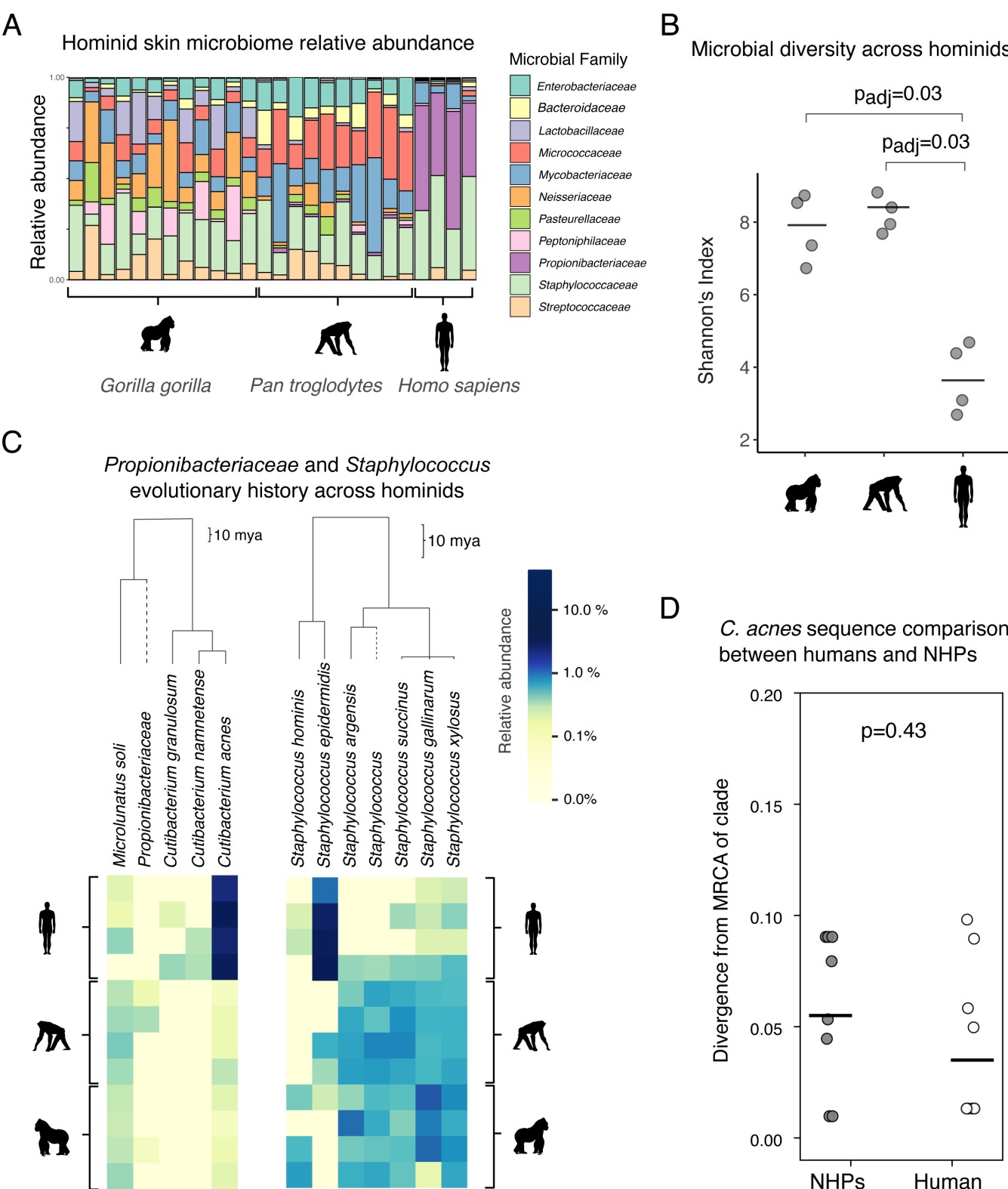

**FIG 1** The facial skin microbiome (FSM) of humans has decreased diversity and is dominated by organisms with no close relatives on other hominids. (A) Relative abundance taxonomic bar plot across all 16S samples, collapsed at the family level. Grey bars at the top denote binned taxa that are less than 1% in relative abundance. (B) At the ASV level, human FSMs are significantly less diverse than chimpanzee and gorilla FSMs (Shannon's Index). Black lines indicate means. Significant comparisons denoted with Bonferroni-corrected $P$ values based on a Kruskal-Wallis test. (C) Phylogenetic analysis indicates that there is no close relative of *C. acnes* on chimpanzee or gorilla FSMs. In contrast, closely related staphylococci are found on all three primates. Heatmap of species abundances from (Continued on next page)

Fig 1 (Continued)

*Propionibacteriaceae* (family level; left) species and *Staphylococcus* (genus level; right) species. Samples were merged by individual host (by summing all sample reads together per individual), rarefied to counts of 10k to normalize sample sequencing depth, and filtered to only include microbes that appear in at least two samples in order to focus on prominent microbes. Scale bars indicate the approximate age of branches on the phylogenies, highlighting the larger depth of the pictured *Propionibacteriaceae* phylogeny. Read abundances are log-transformed to allow for a smoother gradient between low and high read counts. (D) Phylogenomic analysis of the trace amount of *C. acnes* sequences found in non-human primates (NHP) metagenomes suggests *C. acnes* on NHPs in this data set has a direct human origin. We searched for evidence that *C. acnes* on NHPS was significantly diverged from any known *C. acnes* genomes. Divergence was quantified by first defining major intraspecies clades within the *C. acnes* phylogeny and then looking for instances where a NHP metagenome contained some, but not all, of the mutations specific to a known clade, which might suggest a deep branching event (Fig. S1; see the text). *C. acnes* found on NHPs was not more substantially diverged from publicly available genomes than that found on human handlers (*P* value is derived from a two-sided Wilcoxon rank-sum test).

relative. Thus, *C. acnes* likely colonized an already diverged human lineage rather than being conserved across primates through a recent common ancestor.

In terms of the FSM as a whole, gorilla and chimpanzee FSMs were significantly more similar to one another than either were to human FSMs (*P* < 0.001; Fig. 2A; also see Fig. 2S). Thus, despite all hominids sharing similar facial features and the fact that the people worked closely with the apes (in protected contact), the human faces had significantly less diverse and more compositionally distinct microbiota.

At the level of function as determined by metagenomics, human and NHP skin microbiomes had much closer pairwise distances (Fig. 2A; Fig. S3) compared to 16S samples, suggestive of functional redundancy in FSMs across host species. However, we found that human FSMs had a lower relative abundance of fatty acid biosynthesis pathways compared to NHPs (DESeq2, $P_{adj}$ < 0.001, Fig. 2B; Table S4). This decrease in bacterial biosynthesis of lipids is consistent with an increased reliance on host-produced sebaceous lipids, which are produced in exceptionally high amounts on human skin relative to other animals (10). *C. acnes* has many genes for hydrolyzing triglycerides (9), which are then liberated extracellularly for utilization by other species (15). Consistent with a model of increased reliance on host lipid usage in humans, depleted biosynthesis pathways included those for triglycerides found in high abundance on human skin, including palmitic acid (Fig. 2C; Table S5).

Skin microbes provide colonization resistance and interact with the immune system in inflammatory skin processes, including wound healing, acne, and psoriasis (16). The uniqueness of the human FSM revealed here—in terms of diversity, composition, and metabolic pathways—may pose a challenge for animal models which seek to replicate these complex interactions. It is worth noting the small sample size of our study. Future studies with larger sample sizes should conduct more comprehensive samplings of hominid FSMs from wild NHPs and humans from more diverse geographies and lifestyles —ranging from nomadic, rural, and urban—to fully understand the effects of environment and evolution on our skin microbiome composition and function. If *Cutibacterium* is truly specific and universal across humans, a deeper understanding of its biology and evolution may illuminate historical events in human evolution and diversification.

## Sample collection

We sampled the facial skin microbiomes (FSMs) of four chimpanzees, four gorillas, and four humans at Lincoln Park Zoo in Chicago, IL, USA , in January 2020. Zoo staff acclimatized the chimpanzees and gorillas to voluntary, awake sample collection procedures prior to data collection. All facial swabs were collected from the apes in protected contact (i.e., through the animals' enclosure mesh) by using a 9-inch plastic swab holder that extended the distance between the human care staff and the non-human primates (NHPs). We sampled each individual three times: targeting different regions of the face (cheek, chin, and forehead), generating a total of 24 samples across all NHPs. Two swabs, from one chimpanzee individual, did not pass quality control post-sequencing, leaving 22 NHP skin samples for analysis. Because the face site has been studied in humans and found to be homogenous (2, 17), we took single swabs of the human subjects' faces (*n* = 4), generating a total of 26 samples. For each sample, a swab (Puritan, REF#25–1506 1PF

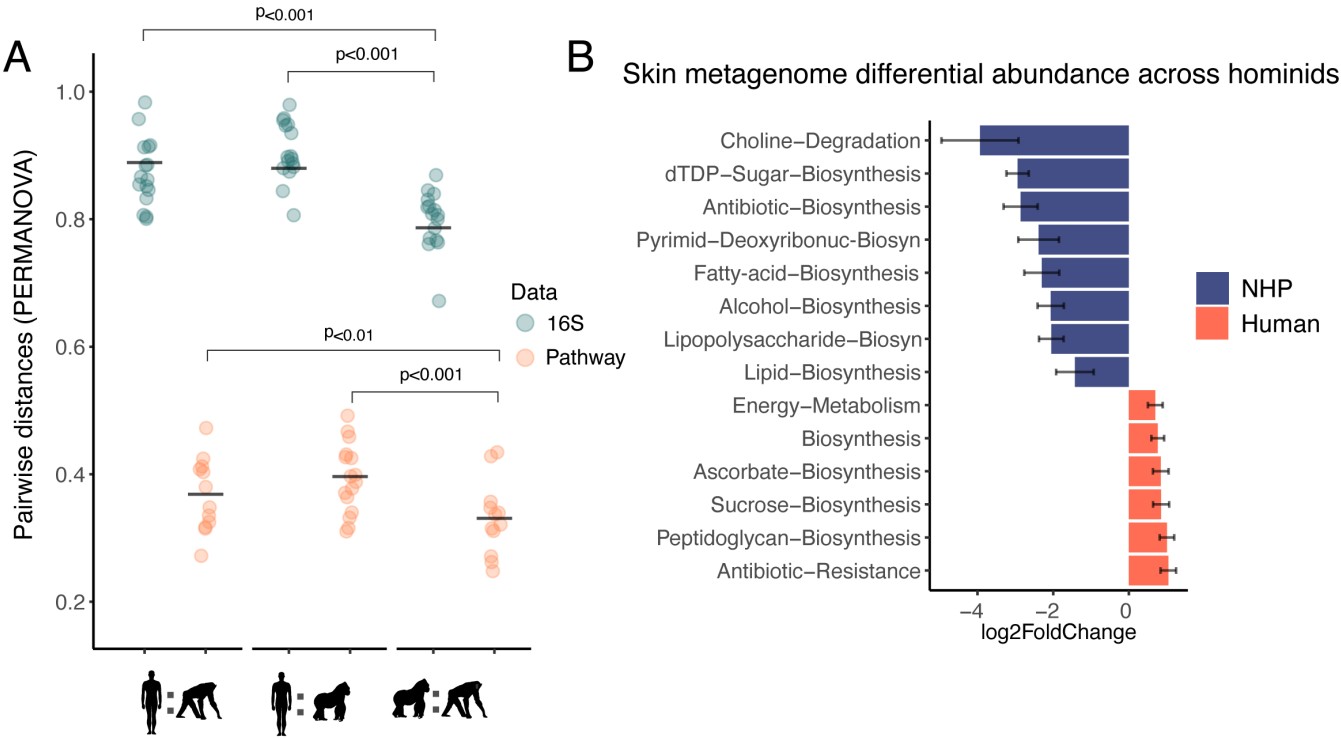

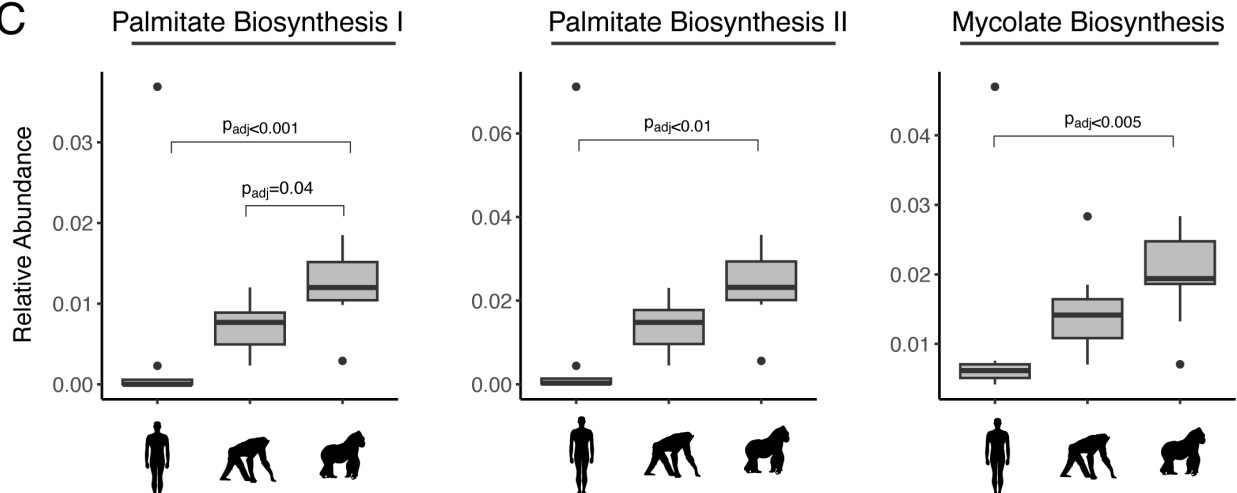

**FIG 2** The human facial skin microbiome (FSM) is distinct and encodes for less lipid biosynthesis, consistent with increased usage of host-derived lipids. (A) A pairwise comparison of Bray-Curtis distances between humans and non-human primates (NHPs) and between gorillas and chimpanzees. *P* values denote significant differences between comparisons. (B) DESeq2 results for the top differentially abundant microbial pathways (MetaCyc) across humans and NHPs. Data are represented as log$_2$ fold changes between the two host groups. (C) The relative abundance of the three most abundant fatty acid pathways across humans, gorillas, and chimpanzees. Palmitate is an abundant lipid on human skin. A Kruskal-Wallis test was used to identify significant differentially abundant fatty acid pathways across hosts, and reported *P* values are Bonferroni-corrected. Metagenomic samples were merged by individual and rarefied to 20,000 reads for A and C.

100) was first dipped into sterile phosphate-buffered saline (PBS) with 0.05% (wt/vol) Tween80, and then applied with light pressure with upward and downward strokes while moving swabs laterally and rotating the swab. The tip of each swab was immediately broken off into 1.5-mL cryotubes containing 1,000 µL of storage solution (20:80 glycerol:PBS with 0.05% [wt/vol] cysteine [as a reducing agent], which had been vacuum

filtered at 0.22 µm and reduced in an anaerobic chamber). Upon completion of a sampling kit, all samples were vortexed for 60 s at maximum speed and placed on dry ice. Samples were transferred to long-term storage in a −80°C freezer the same day they were collected.

## 16S processing

DNA was extracted from 100 µL of sample using the PURElink Genomic DNA Purification Kit (Invitrogen, K182002) following the protocol for gram-positive bacterial cells. The following adjustments were made to the protocol to improve the lysis of difficult-to-lyse organisms, including *C. acnes*: the lysozyme treatment was increased to overnight incubation (16 h) at 37°C, and the incubation with proteinase K was increased to 3 h at 55°C. We also extracted four negative controls and processed them through sequencing to assess for contamination. The V1-V3 region of the bacterial 16S ribosomal RNA gene was amplified using Kapa HIFI HotStart Readymix (Roche, 07958935001) and 27F-plex and 534R-plex primers (Table S7). PCR of samples was performed as follows: 3 min of 95°C; 36 cycles of 98°C for 30 s, 54.5°C for 30 s, and 72°C for 30 s; and 72°C for 5 min. Samples were cleaned following the protocol outlined in the Illumina 16S Metagenomic Sequencing Library Preparation Guide (link). Amplicon product was validated by TapeStation (Agilent) and quantified by SYBR Safe (Thermo, #S33102). Samples were indexed with standard Illumina primers and cleaned again following the Illumina guide. Indexing PCR was performed as follows: 3 min of 95°C; eight cycles of 95°C for 30 s, 55°C for 30 s, and 72°C for 30 s; and 72°C for 5 min. Samples were pooled in equimolar ratios after validation of amplicon size by TapeStation (Agilent) and quantification by SYBR Safe (Thermo, #S33102). Pooled samples were sequenced with the Illumina MiSeq platform using the v2 chemistry for 250-bp paired-end reads at the MIT BioMicroCenter.

## Metagenomic processing

Genomic libraries for metagenomic samples were processed as previously described (2). Lysates were produced with an overnight lysozyme treatment at 37°C followed by a 3-h proteinase K digestion at 55°C and purified with Purelink gDNA extraction kits (Thermo, #K182104A). Libraries were prepared using a miniaturized protocol of Hackflex (18 ) where sample input was reduced to 10 µL at 1 ng/µL at the tagmentation step, and the tagmentation stop step was omitted. Standard Illumina primers and KAPA HiFi Master Mix (Roche, 07958935001) were used to index samples with the following PCR protocol: 72°C for 3 min; 98°C for 5 min; 19 cycles of 98°C for 10 s, 62°C for 30 s, and 72°C for 30 s; and 72°C for 5 min. Samples were cleaned, pooled, and sequenced using 150-bp paired-end reads on the Illumina NextSeq500.

## Data processing

We processed the 16S sequences using QIIME 2 (v. 2023.7) using the microbiome data science platform (19) for quality control, ASV taxonomy assignment, and community diversity analyses. We demultiplexed and denoised the sequencing data using DADA2 (20) and merged the resulting output into a feature table for subsequent analysis. We assigned taxonomy to ASVs, using a naïve Bayes taxonomy classifier trained on the greengenes2 database (21), conducting reference sequence clustering at 99% similarity. To avoid unwanted reads, we removed singletons, and for analyses that involved relative abundance, we removed reads that only showed up in one sample. To ensure that microbiomes only included microbial sequences, we removed any ASVs assigned to eukaryotes or chloroplasts.

To trim adapters and low-quality bases from the metagenomic sequences and remove host contamination, we used the kneaddata pipeline (v0.11), which includes Trimmomatic (22) and Bowtie 2 (23). To add taxonomy, we mapped reads to the Web of Life database (24), and for functional annotations, we used MetaCyc and the Kyoto Encyclopedia of Genes and Genomes (KEGG) databases. Because we saw a stronger

bias toward human-associated pathways with the KEGG database, we used the MetaCyc database for a majority of differential analyses between human and NHP skin microbiomes. To bin genes into functional categories, we used Woltka's collapse function to generate functional types.

In total, 16S sequencing of 26 skin samples (two samples were removed due to low sequencing depth) generated 1,409,637 reads after denoising, belonging to 19,737 ASVs. Samples had a median of 48,138 reads and a minimum of 19,368 reads. Shotgun sequencing generated 3,053,249 reads after trimming and host-filtering. Samples had a median of 81,638 reads with a minimum of 6,502. Using the MetaCyc database, 4,296 pathways were identified, while the KEGG database matched to 6,002 pathways.

## Statistical analysis

To conduct beta diversity analyses, we utilized QIIME 2's beta diversity functions on both our 16S amplicon sequence variant (ASV) table of reads and a metagenomic gene table to analyze diversity and function in parallel. We rarefied reads to 10,000 in order to retain 95% of the samples before conducting beta diversity analyses for 16S and 20,000 reads to retain 100% of the metagenomic samples. We based the principal coordinate analysis (PCoA) plots and permutational multivariate analysis of variance (25) analyses on the taxonomic OTU table and a Bray-Curtis (26) distance matrix. For the metagenomic table, we also relied on the Bray-Curtis distance matrix. We chose the Bray-Curtis metric as we could standardize this across both sequencing types (in lieu of UniFrac, e.g., which is specific to 16S data), while also taking into account relative abundance for each data set. To compare alpha diversities, we opted for Shannon's Index (27) since this index was comparable across both 16S and metagenomic data sets. We also supplemented this with observed read counts to ensure our results were robust across both diversity metrics. Differences in alpha diversity across hosts were determined with a Kruskal-Wallis test (28) on Bonferroni-corrected $P$ values.

Because we found no differences across facial sites in NHP FSMs, we merged samples per individual for a majority of our analyses. We did this to better equalize sample sizes across hosts, and we accounted for sequencing depth by rarefying reads to 10,000 for 16S samples and 20,000 for metagenomic samples.

To analyze differentially abundant reads across host taxonomy, we employed DESeq2 to quantify differentially abundant microbes and pathways. We also performed DESeq2 on the genes specific to bacterial taxa by utilizing a gene count table stratified with taxonomy—made with Woltka's stratifying function—to investigate certain bacteria of interest and whether their gene counts differed across hosts. We used raw, minimally filtered (i.e., filtered for Eukaryota, Chloroplast, or unassigned reads), sequence tables for all differential analyses, following author recommendations (29). For visualizing differential abundance, we filtered the results by $P$ value (which directly correlates with and showed the top 14 hits with the smallest $P$ values). If fewer than 14 taxa or pathways were significant ($P_{adj} > 0.05$), we showed all significant hits.

To visualize *Cutibacterium* and *Staphylococcus* diversity across hosts, we constructed a heatmap of the most abundant taxa within each genus. To focus on both the most abundant and prevalent taxa, we rarefied the 16S table to 10,000 reads per sample and excluded microbes that only appeared in one subject. We also constructed a similar heatmap of *Propionibacteriaceae* spp. (family level) to visualize the abundance of *Cutibacterium* relatives across hosts. We used QIIME 2's heatmap plugin that uses the EMPress package (30) that allows for phylogenetically aware visualizations of microbiome data. To visualize the relationships between samples and microbes, we employed a cluster analysis based on Euclidean distances to annotate samples and microbes with dendrogram connections. To visualize all phylogenetic relationships, we also constructed a phylogeny of all identified microbes in the data utilizing our taxonomy table and a rooted tree made with MAFFT (v7), and divergence times between microbes were calculated with TimeTree 5 (timetree.org). For our analyses based on taxonomic identification rather than function, we opted to use the 16S data over the metagenomic

data to minimize false-positive discovery associated with metagenomic profiling (31, 32). We also verified our 16S taxonomic identifications with BLAST and were able to confirm the identities of taxa shown in Fig. 1C. When there was a discrepancy in species identification, we opted for the BLAST result over the Greengenes result.

To determine whether the *C. acnes* found in NHPs were native to NHP skin or had a human origin, we conducted a phylogenomic analysis of the *C. acnes* metagenomic sequences found on NHP samples (Fig. S4). We reasoned that *C. acnes* that is truly native to NHPs should be diverged to some extent from *C. acnes* found on humans and therefore used an early version of PHLAME, a new reference-based metagenomic strain profiler that can identify the presence of novel strains (33). First, we generated maximum likelihood core-genome phylogeny of *C. acnes* using 358 public reference genomes isolated from humans and identified major intraspecies clades. We searched for the presence of each major clade in metagenomes by identifying clade-specific SNPs, which are uniquely and unanimously shared by all members of the clade. We then looked for cases where a subset of SNPs specific to a clade was systematically missing, which is evidence of a novel strain that diverged substantially before the common ancestor of that clade. We quantified this divergence by estimating the number of SNPs that were systematically missing over the total number of SNPs specific to that clade (bounded between 0 and 1). We then employed a two-sided Wilcoxon rank-sum test comparing *C. acnes* divergence between NHP and handler skin microbiome samples across clades as well as with the clades combined.

To look for evidence of *C. acnes* relatives in metagenome-assembled genomes, we assembled contigs directly from NHP metagenomes using metaSPAdes (v3.13). To identify contigs that could reasonably belong to *C. acnes* or a close relative, we used BLAST (v2.7.1) to query all contigs against a database consisting of 24 representative *C. acnes* and *Cutibacterium granulosum* genomes. For any contig that returned a hit, we used BLAST again against the National Center for Biotechnology Information nucleotide database and selected only contigs with a top hit belonging to *C. acnes*. We recovered only five contigs after this second round of filtering, which all had either alignment lengths or percent IDs that were too low to reasonably originate from *C. acnes*.

To analyze differences in fatty acid pathways across hosts, we employed a Kruskal-Wallis test on Palmitate 1, Palmitate 2, and Mycolate biosynthesis pathways across humans and NHPs. These three pathways were identified with the MetaCyc database and were the three most relatively abundant pathways across all samples.

## Culturing attempts

We attempted to isolate *Cutibacterium* spp. directly from primate skin swabs. Following an existing protocol (34) for isolating *Cutibacterium* species from human skin, we streaked from forehead and cheek swabs of each primate (eight total) directly onto reinforced clostridial medium (RCM) agar plates, followed by anaerobic incubation at 33°C for 5 days. We only recovered colonies from two gorillas and two chimpanzees. We picked 38 colonies with *Cutibacterium*-like morphologies (focusing on pale, shiny, and circular colonies, while ignoring yellow, matte, and irregular colonies). We PCR amplified the 16S rRNA gene from each colony with 27F-1492R primers, using 35 cycles of 94°C for 30 s, 54.5°C for 30 s, and then 72°C for 30 s. Amplicons were Sanger sequenced before classifying against the SILVA small ribosomal subunit database (v138.1) for taxonomic identification. Among the 38 colonies we sequenced, 25 were identified as *Staphylococcus*, one as *Enterococcus*, 10 were unclassified, and two had evidence of mixed peaks and were not considered (Table S9).,,

## ACKNOWLEDGMENTS

We would like to thank Lincoln Park Zoo for providing permissions for this study, the BioMicroCenter at MIT for assistance in DNA sequencing, and especially Stephen Ross and Jill Moyse for their logistical support for all data collection. This work was supported in part by NIH grant (DP2OD028909) to T.D.L.

## AUTHOR AFFILIATIONS

[1]Department of Anthropology, Northwestern University, Evanston, Illinois, USA

[2]Department of Anthropology, University of Texas at San Antonio, San Antonio, Texas, USA

[3]Institute for Medical Engineering and Science, Massachusetts Institute of Technology, Cambridge, Massachusetts, USA

[4]Department of Civil and Environmental Engineering, Massachusetts Institute of Technology, Cambridge, Massachusetts, USA

[5]Department of Molecular and Comparative Pathobiology, Johns Hopkins University School of Medicine, Baltimore, Maryland, USA

## AUTHOR ORCIDs

Samuel Degregori  http://orcid.org/0000-0002-4616-580X

Katherine R. Amato  http://orcid.org/0000-0003-2722-9414

Tami D. Lieberman  http://orcid.org/0000-0001-5430-3937

## FUNDING

| Funder | Grant(s) | Author(s) |
| --- | --- | --- |
| National Institutes of Health | DP2OD028909 | Tami D. Lieberman |

## DATA AVAILABILITY

All source code and data used for analysis is available at https://github.com/samd1993/SkinProject. The source code for our metagenomic strain profiler, PHLAME, is available at https://github.com/quevan/phlame. Sequences for both 16S and shotgun sequencing, along with associated metadata files and a STORMS checklist, are available at https://figshare.com/articles/dataset/Sequences_MG_and_16S/28135763. Sequences are also deposited in the SRA database under BioProject accession PRJNA1209946.

## ETHICS APPROVAL

Samples were collected under IRB protocol STU00206091.

## ADDITIONAL FILES

The following material is available online.

### Supplemental Material

**Supplemental material (mSystems00081-25-s0001.pdf).** Supplemental figures and tables.

### Open Peer Review

**PEER REVIEW HISTORY (review-history.pdf).** An accounting of the reviewer comments and feedback.

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
