## [Reviewer comments · mSystems]

The microbiome of the human facial skin is unique compared to that of other hominids

Sam Degregori, Melissa Manus, Evan Qu, Calen Mendall, Jacob Baker, Lydia Hopper, Katherine R. Amato, and Tami Lieberman

Corresponding Author(s): Tami Lieberman, Massachusetts Institute of Technology

Review Timeline:

Submission Date:	January 16, 2025
Editorial Decision:	March 19, 2025
Revision Received:	April 8, 2025
Accepted:	April 24, 2025

Editor: Sarah Hird

Reviewer(s): Disclosure of reviewer identity is with reference to reviewer comments included in decision letter(s). The following individuals involved in review of your submission have agreed to reveal their identity: Holger Brüggemann (Reviewer #2)

Transaction Report:

DOI: <https://doi.org/10.1128/msystems.00081-25>

Hi Tami -

The reviewers and I agree this is a nice manuscript that is very close to publication. Please address all the reviewer comments and I'd like to second the comment from reviewer 2 about being a little more explicit about the limitations of the study. Let me know if you have any questions - thanks for submitting to mSystems!

Sarah

PS - Form letter with all the details below...

Re: mSystems00081-25 (The microbiome of the human facial skin is unique compared to that of other hominids)

Dear Dr. Tami Lieberman:

Revision Guidelines

Sincerely,
Sarah Hird
Editor

Reviewer #1 (Comments for the Author):

The manuscript "The microbiome of the human facial skin is unique compared to that of other hominids" by S. Degregori et. al. is a dynamic view of the facial skin microbiome comparing primates. This study utilizes both genomic 16S amplicon and metagenomic DNA sequencing as well as phylogenetic diversity of microbes, particularly *Cutibacterium acnes*. In addition, this study looks deeper to understand functionality of these facial microbiomes to distinguish the human facial skin microbiome biosynthesis from those of the closest primate relatives. Overall, this study was well thought-out and went deeper to answer questions in addition to the initial scope. However, increasing the number of individuals sampled would be advantageous to the statistical power of the analyses. Future studies could encompass several Zoo sites of gorillas and chimpanzees and include humans not exposed to the animals tested. The results of this study are helpful to understand phylogenomic origin of *C. acnes* on human facial skin and future studies on human skin health.

Minor corrections:

1. Line 191-195: The number of samples collected is unclear: (a) the total number being 22 samples, it should be stated that one *Pan troglodytes* had only one swab (b) 22 is for only non-human primates, not all individuals in this sentence.
2. Differentiating the main text sections into Introduction, Results and Discussion, and Conclusion would be more helpful to the traditional reader to follow the scientific experimental logic and research paper layout with perhaps Results and Discussion subheadings describing the main findings.
3. Specifically stating the hypothesis to the questions raised at the end of the introduction will direct the intention of what was to be found in this study.
4. Lines 76, 149, 179, 470, & 689: Change "Krusk-Wallis" to "Kruskal-Wallis" test.
5. A few references are missing, such as Kruskal-Wallis, Shannon, and Biom files used in Qiime2, Bray-Curtis, Permanova.
6. The methods are well written and detailed. However, there is a minor exclusion of Kruskal-Wallis used in Bonferroni-corrected p-values.
7. Lines 75-76: Figure 1B is described as family level, but the caption for Fig. 1B states the analysis on the species level. Please explain or correct accordingly.
8. Line 330: Change "alonog" to "along".
9. Line 85: Change "padj<.04" to "padj<0.04"
10. There were individuals sampled under the type "Acera" in the metadata file. Please explain what those human subjects were in the methods.
11. In table S7, two sets of 16S primers are said to be used, which contrasts with the methods text that only V1-V3 set was used. Please remove the V3-V4 primer set from the table or explain their use in the methods text.

Reviewer #2 (Comments for the Author):

This is an interesting but small sample size study.

major comment:

the authors should include the limitations of this study: very small sample size, no viability tests (bacterial cultivation), only one sampling method (swabs)

minor comments

55

It should be noted that this is based on seq data not CFU; seq data very likely overestimates viable bacteria count

68

It should be noted that *C. acnes* is not very abundant in young humans, its population expands later (early teenager)

80

It would have been interesting to look at Staph species/strains (see also lines 97etc)

86, 90, 99

Wouldn't cultivation of *C. acnes* on specific agar and subsequent WGS be the most appropriate analysis? Most likely these reads are contamination (not from viable *C. acnes*)

Reviewer comments:

The manuscript **“The microbiome of the human facial skin is unique compared to that of other hominids”** by S. Degregori et. al. is a dynamic view of the facial microbiome comparing primates. This study utilizes both genomic 16S amplicon and metagenomic DNA sequencing as well as phylogenetic diversity of microbes, particularly *Cutibacterium acnes*. In addition, this study looks deeper to understand functionality of these facial microbiomes to distinguish the human facial skin microbiome biosynthesis from those of the closest primate relatives. Overall, this study was well thought-out and went deeper to answer questions in addition to the initial scope. However, increasing the number of individuals sampled would be advantageous to the statistical power of the analyses. Future studies could encompass several Zoo sites of gorillas and chimpanzees and include humans not exposed to the animals tested. The results of this study are helpful to understand phylogenomic origin of *C. acnes* on human and future studies on human skin health.

Minor corrections:

1. Line 191-195: The number of samples collected is unclear: (a) the total number being 22 samples, it should be stated that one *Pan troglodytes* had only one swab (b) 22 is for only non-human primates, not all individuals in this sentence.
2. Differentiating the main text sections into Introduction, Results and Discussion, and Conclusion would be more helpful to the traditional reader to follow the scientific experimental logic and research paper layout with perhaps Results and Discussion subheadings describing the main findings.
3. Specifically stating the hypothesis to the questions raised at the end of the introduction will direct the intention of what was to be found in this study.
4. Lines 76, 149, 179, 470, & 689: Change “Krusk-Wallis” to “Krustal-Wallis” test.
5. A few references are missing, such as Kruskal-Wallis, Shannon, and Biom files used in Qiime2, Bray-Curtis, Permanova.
6. The methods are well written and detailed. However, there is a minor exclusion of Kruskal-Wallis used in Bonferroni-corrected p-values.
7. Lines 75-76: Figure 1B is described as family level, but the caption for Fig. 1B states the analysis on the species level. Please explain or correct accordingly.
8. Line 330: Change “alonog” to “along”.
9. Line 85: Change “ $\text{padj} < .04$ ” to “ $\text{padj} < 0.04$ ”
10. There were individuals sampled under the type “Acera” in the metadata file. Please explain what those human subjects were in the methods.

11. In table S7, two sets of 16S primers are said to be used, which contrasts with the methods text that only V1-V3 set was used. Please remove the V3-V4 primer set from the table or explain their use in the methods text.

Dear Editor and associates,

We appreciate the constructive feedback on our manuscript titled, “The microbiome of the human facial skin is unique compared to that of other hominids”. We agree that some methodological details can be made clearer, including the small sample size and some missing details on our methodology choices. Below we have outlined all the changes we made per each reviewer comment. We hope these will suffice for publication. Our written responses are colored in blue for your convenience, and in-line quoted changes are bolded.

Reviewer #1 (Comments for the Author):

The manuscript "The microbiome of the human facial skin is unique compared to that of other hominids" by S. Degregori et. al. is a dynamic view of the facial skin microbiome comparing primates. This study utilizes both genomic 16S amplicon and metagenomic DNA sequencing as well as phylogenetic diversity of microbes, particularly *Cutibacterium acnes*. In addition, this study looks deeper to understand functionality of these facial microbiomes to distinguish the human facial skin microbiome biosynthesis from those of the closest primate relatives. Overall, this study was well thought-out and went deeper to answer questions in addition to the initial scope. However, increasing the number of individuals sampled would be advantageous to the statistical power of the analyses. Future studies could encompass several Zoo sites of gorillas and chimpanzees and include humans not exposed to the animals tested. The results of this study are helpful to understand phylogenomic origin of *C. acnes* on human facial skin and future studies on human skin health.

Minor corrections:

1. **Line 191-195:** The number of samples collected is unclear: (a) the total number being 22 samples, it should be stated that one Pan troglodytes had only one swab (b) 22 is for only non-human primates, not all individuals in this sentence.

Thank you for catching these important details. Two swabs did not pass quality control post sequencing from a chimpanzee individual and we have clarified this in the main text. **Line 193-195 now reads:**

“We sampled each individual three times: targeting different regions of the face (cheek, chin, and forehead), generating a total of 24 samples across all non-human primates (NHPs). Two swabs, from one chimpanzee individual, did not pass quality control post sequencing, leaving 22 NHP skin samples for analysis.”

2. Differentiating the main text sections into Introduction, Results and Discussion, and Conclusion would be more helpful to the traditional reader to follow the scientific experimental logic and research paper layout with perhaps Results and Discussion subheadings describing the main findings.

Thank you for this suggestion. However, our article is submitted as an Observation format, which requires that we stick to the abridged formatting requirements laid out by *mSystems*.

3. Specifically stating the hypothesis to the questions raised at the end of the introduction will direct the intention of what was to be found in this study.

We agree this will help direct the reader towards the study's aim.
Lines 71-74 now read:

“Here, to understand if *C. acnes* has a close relative on related hominids, and to more generally understand how the human facial skin microbiome differs from other hominids, we profiled the facial skin microbiome (FSM) of 4 chimpanzees and 4 gorillas at Lincoln Park Zoo...”

4. Lines 76, 149, 179, 470, & 689: Change "Krusk-Wallis" to "Krustal-Wallis" test.

Fixed, thank you.

5. A few references are missing, such as Kruskal-Wallis, Shannon, and Biom files used in Qiime2, Bray-Curtis, Permanova.

We have added all references for each analysis.

6. The methods are well written and detailed. However, there is a minor exclusion of Kruskal-Wallis used in Bonferroni-corrected p-values.

Thank you for catching this. Lines 274-276 now read:

“Differences in alpha diversity across hosts were determined with a Kruskal-Wallis test (Theodorsson-Norheim, 1986) on Bonferroni-corrected p-values.”

And lines 331-334 now read:

“To analyze differences in fatty-acid pathways across hosts, we employed a Krusk-Wallis test on Palmitate 1, Palmitate 2, and Mycolate biosynthesis pathways across humans and NHPs. These three pathways were identified with the Metacyc database and were the three most relatively abundant pathways across all samples.”

7. Lines 75-76: Figure 1B is described as family level, but the caption for Fig. 1B states the analysis on the species level. Please explain or correct accordingly.

Thank you for catching this. We have corrected the main text to correctly state that the diversity differences in Fig 1B are at the ASV level. Lines 77-81 now read:

“Family level composition is shown in Fig. 1A. Human FSMs exhibited significantly less alpha diversity, at the ASV level, than gorilla and chimpanzee FSMs ($p_{adj} < 0.05$ Kruskal-Wallis; Fig. 1B), echoing results from other body sites (Council et al., 2016).”

This is also reflected in the Figure 1B caption as well.

“B) At the ASV level, human FSMs are significantly less diverse than chimpanzee and gorilla FSMs (Shannon’s Index).”

8. Line 330: Change "alonog" to "along".

Fixed. Thank you.

9. Line 85: Change "padj<.04" to "padj<0.04"

Fixed. Thank you.

10. There were individuals sampled under the type "Acera" in the metadata file. Please explain what those human subjects were in the methods.

These samples were incorrectly placed as they were not analyzed as part of this study. We apologize for the confusion and these will be removed.

11. In table S7, two sets of 16S primers are said to be used, which contrasts with the methods text that only V1-V3 set was used. Please remove the V3-V4 primer set from the table or explain their use in the methods text.

Thank you for catching this. The V3-V4 primer set was not used in this study and is now removed from the table.

~~~~~

Reviewer #2 (Comments for the Author):

This is an interesting but small sample size study.

major comment:

the authors should include the limitations of this study: very small sample size, no viability tests (bacterial cultivation), only one sampling method (swabs)

We appreciate the constructive feedback. We agree that these limitations are important to note and as such we have added further details in the text to make this clearer. As shown below, we have added details of cultivation efforts in the lab that we have now incorporated into the study. And to address the small sample size 135-140 now read:

“It is worth noting the small sample size of our study. Future studies with larger sample sizes should conduct more comprehensive samplings...”

minor comments

55

It should be noted that this is based on seq data not CFU; seq data very likely overestimates viable bacteria count

While *C. acnes* is much more viable in skin pores than on the surface, the relative abundance of viable CFUs on the skin is also similar. We have now added a citation to Noble and Sommerville 1981 which finds similar results. Line 55 now reads:

**“...and constitute ~70% of the relative abundance of facial skin microbes (based on sequencing data) of adults...”**

68

It should be noted that *C. acnes* is not very abundant in young humans, its population expands later (early teenager)

Given the short and focused nature of this Observation, we have chosen not to include this interesting fact. We note that we mention adults specifically when discussing observations of *C. acnes* dominance.

80

It would have been interesting to look at Staph species/strains (see also lines 97etc)

As shown in Fig 1C, several species of Staph are quite abundant and diversified across primates in comparison to *Cutibacterium spp.*

86, 90, 99

Wouldn't cultivation of *C. acnes* on specific agar and subsequent WGS be the most appropriate analysis? Most likely these reads are contamination (not from viable *C. acnes*)

We agree. We did, in fact, attempt to cultivate *C. acnes* in agar from primate skin swabs and were unsuccessful in doing so. We now mention this in the text, have added a supplemental table (Table S9), and described our methods at the end of the methods section.

Lines 111-113 now read:

**“Moreover, attempts to culture *C. acnes* from these NHP samples, were unsuccessful (Methods, Supplementary Table S9).”**

Lines 337-348 now read:

**“We attempted to isolate *Cutibacterium spp.* directly from primate skin swabs. Following an existing protocol (Conwill et al., 2022) for isolating *Cutibacterium* species from human skin, we streaked from forehead and cheek swabs of each primate (eight total) directly onto RCM agar plates, followed by anaerobic incubation at 33C for five days. We only recovered colonies from two gorillas and two chimpanzees. We picked 38 colonies with *Cutibacterium*-like morphologies (focusing on pale, shiny and circular colonies, while ignoring yellow, matte, and irregular colonies). We PCR amplified the 16S rRNA gene**

from each colony with 27F-1492R primers, using 35 cycles of 94C for 30s, 54.5C for 30s, then 72C for 30s. Amplicons were Sanger sequenced before classifying against the SILVA small ribosomal subunit database (v138.1) for taxonomic identification. Among the 38 colonies we sequenced, 25 were identified as *Staphylococcus*, 1 as *Enterococcus*, 10 were unclassified, and two had evidence of mixed peaks and were not considered (Table S9).”

Thank you for reviewing our paper so thoroughly.

Sincerely,  
Tami Lieberman

Hi Tami -

Thanks for the revisions. The reviewers were satisfied and I'm happy to accept the paper. I have two small things to draw attention (but that I don't want to impede the progression of the paper).

1. We noticed that you identify humans by their first names in the metadata and since the zoo is also named, it would be relatively easy (maybe) for someone to tie the sequence data to a person. Do you know if their informed consent is required before publishing? This is probably something I should know, so I'm happy to look into it at mSystems, if we need to. (Sorry for my ignorance, I don't work with humans!) Feel free to reach out to me as needed (sarah.hird@uconn.edu).
2. Table S9 has a bunch of typos in the taxonomic names - Staphylococcus and Enterococcus are missing letters. At least I think that's the issue. Worth you checking, just to be sure.

Thank you (form letter with details below) -  
Sarah

Re: mSystems00081-25R1 (The microbiome of the human facial skin is unique compared to that of other hominids)

Dear Dr. Tami Lieberman:

Your manuscript has been accepted, and I am forwarding it to the ASM production staff for publication. Your paper will first be checked to make sure all elements meet the technical requirements. ASM staff will contact you if anything needs to be revised before copyediting and production can begin. Otherwise, you will be notified when your proofs are ready to be viewed.

**Data Availability:** ASM policy requires that data be available to the public upon online posting of the article, so please verify all links to sequence records, if present, and make sure that each number retrieves the full record of the data. If a new accession number is not linked or a link is broken, provide production staff with the correct URL for the record. If the accession numbers for new data are not publicly accessible before the expected online posting of the article, publication may be delayed; please contact ASM production staff immediately with the expected release date.

**Publication Fees:** For information on publication fees and which article types have charges, please visit our website. We have partnered with Copyright Clearance Center (CCC) to collect author charges. If fees apply to your paper, you will receive a message from no-reply@copyright.com with further instructions. For questions related to paying charges through RightsLink, please contact CCC at ASM\_Support@copyright.com or toll free at +1-877-622-5543. CCC makes every attempt to respond to all emails within 24 hours.

**ASM Membership:** Corresponding authors may join or renew ASM membership to obtain discounts on publication fees. Need to upgrade your membership level? Please contact Customer Service at [Service@asmusa.org](mailto:Service@asmusa.org).

**PubMed Central:** ASM deposits all mSystems articles in PubMed Central and international PubMed Central-like repositories immediately after publication. Thus, your article is automatically in compliance with the NIH access mandate. If your work was supported by a funding agency that has public access requirements like those of the NIH (e.g., the Wellcome Trust), you may post your article in a similar public access site, but we ask that you specify that the release date be no earlier than the date of publication on the mSystems website.

**Embargo Policy:** A press release may be issued as soon as the manuscript is posted on the mSystems Latest Articles webpage. The corresponding author will receive an email with the subject line "ASM Journals Author Services Notification" when the article is available online.

**Cover Image Submissions:** If you would like to submit a potential Cover Image, please email a file and a short legend to [mssystems@asmusa.org](mailto:mssystems@asmusa.org). Please note that we can only consider images that (i) the authors created or own and (ii) have not been previously published. By submitting, you agree that the image can be used under the same terms as the published article. Image File requirements: TIF/EPS, 7.5 inches wide by 8.25 inches tall (at least 2,250 pixels wide by 2,475 pixels tall), minimum 300 dpi resolution (600 dpi preferred), RGB, and no figure elements, e.g., arrows or panel labels. The legend should be a short description of the image, 1-2 sentences recommended. Please download and use this interactive template in Adobe to ensure that your proposed cover image meets our size requirements (<https://journals.asm.org/pb-assets/pdf-text-excel-files/ASM-Interactive-Sizing-Cover-Template-1715689791.pdf>).

**Author Video:** For mSystems research articles, you are welcome to submit a short author video for your recently accepted paper. Videos are normally 1 minute long and are a great opportunity for junior authors to get greater exposure. Importantly, this video will not hold up the publication of your paper and you can submit it at any time.

We recognize that the video files can become quite large, so to avoid quality loss ASM suggests sending the video file via <https://www.wetransfer.com/>. When you have a final version of the video and the still ready to share, please send it to mSystems staff at [mSystems@asmusa.org](mailto:mSystems@asmusa.org).

Sincerely,  
Sarah Hird  
Editor  
mSystems

Reviewer #1 (Comments for the Author):

N/A

Reviewer #2 (Comments for the Author):

The authors addressed the issues raised by the reviewers